Interpopulational differences in the nutritional condition of Aequiyoldia eightsii (Protobranchia: Nuculanidae) from the Western Antarctic Peninsula during austral summer

Bascur Miguel 1 2 mbascur@magister.ucsc.cl
http://orcid.org/0000-0002-7761-660X Morley Simon A. 3
Meredith Michael P. 3
http://orcid.org/0000-0003-1348-5476 Muñoz-Ramírez Carlos P. 4
Barnes David K. A. 3
http://orcid.org/0000-0002-5917-8925 Schloss Irene R. 5 6 7
http://orcid.org/0000-0003-1028-0328 Sands Chester J. 3
http://orcid.org/0000-0003-2359-4131 Schofield Oscar 8
Román-Gonzaléz Alejandro 9
Cárdenas Leyla 10 11
Venables Hugh 3
Brante Antonio 1 12
Urzúa Ángel 1 12
1 Departamento de Ecología, Facultad de Ciencias, Universidad Católica de la Santísima Concepción , Concepción , Chile
2 Programa de Magister en Ecología Marina, Universidad Católica de la Santísima Concepción , Concepción , Chile
3 British Antarctic Survey, Natural Environment Research Council , Cambridge , United Kingdom
4 Instituto de Entomología, Universidad Metropolitana de Ciencias de la Educación , Santiago , Chile
5 Instituto Antártico Argentino , Buenos Aires , Argentina
6 Centro Austral de Investigaciones Científicas (CADIC-CONICET) , Ushuaia , Argentina
7 Universidad Nacional de Tierra del Fuego , Ushuaia , Argentina
8 Center for Ocean Observing Leadership, Department of Marine and Coastal Sciences, School of Environmental and Biological Sciences, Rutgers University , New Brunswick , United States
9 College of Life and Environmental Sciences, University of Exeter , Cornwall , United Kingdom
10 Centro FONDAP de Investigación en Dinámica de Ecosistemas Marinos de Altas Latitudes (IDEAL) , Valdivia , Chile
11 Instituto de Ciencias Ambientales y Evolutivas, Facultad de Ciencias, Universidad Austral de Chile , Valdivia , Chile
12 Centro de Investigación en Biodiversidad y Ambientes Sustentables (CIBAS), Universidad Católica de la Santísima Concepción , Concepción , Chile
Figuerola Blanca
Electronic publication date: 2021 Dec 21
Publication date: 2021
Volume: 9
Electronic Location ID: e12679
Received 2021 Aug 20; Accepted 2021 Dec 2
Copyright: © 2021 Bascur et al.
Copyright year: 2021
Copyright holder: Bascur et al.
License: This is an open access article distributed under the terms of the Creative Commons Attribution License, which permits unrestricted use, distribution, reproduction and adaptation in any medium and for any purpose provided that it is properly attributed. For attribution, the original author(s), title, publication source (PeerJ) and either DOI or URL of the article must be cited.
License URL: https://creativecommons.org/licenses/by/4.0/

Keywords: Bivalve, Infaunal, Invertebrate, Fatty acid, Physiology

Funding: Comisión Nacional de Investigación Científica y Tecnológica (CONICYT) Natural Environment Research Council (NERC) (ICEBERGS) PII20150078 Fondo Nacional de Desarrollo Científico y Tecnológico (CONICYT- FONDECYT) 3180331 Magister en Ecología Marina and Dirección de Postgrados UCSC Financial support was provided by the Comisión Nacional de Investigación Científica y Tecnológica (CONICYT)- Natural Environment Research Council (NERC) to Antonio Brante, Ángel Urzúa, Alejandro Román-González, Michael P Meredith, David KA Barnes (ICEBERGS project N° PII20150078) and Fondo Nacional de Desarrollo Científico y Tecnológico (CONICYT- FONDECYT) to Carlos Muñoz-Ramírez (project N° 3180331). Miguel Bascur thanks the fellowships granted by the Magister en Ecología Marina and Dirección de Postgrados UCSC. There was no additional external funding received for this study. The funders had no role in study design, data collection and analysis, decision to publish, or preparation of the manuscript.

==============================
The Western Antarctic Peninsula (WAP) is a hotspot for environmental change and has a strong environmental gradient from North to South. Here, for the first time we used adult individuals of the bivalve Aequiyoldia eightsii to evaluate large-scale spatial variation in the biochemical composition (measured as lipid, protein and fatty acids) and energy content, as a proxy for nutritional condition, of three populations along the WAP: O’Higgins Research Station in the north (63.3°S), Yelcho Research Station in mid-WAP (64.9°S) and Rothera Research Station further south (67.6°S). The results reveal significantly higher quantities of lipids (L), proteins (P), energy (E) and total fatty acids (FA) in the northern population (O’Higgins) (L: 8.33 ± 1.32%; P: 22.34 ± 3.16%; E: 171.53 ± 17.70 Joules; FA: 16.33 ± 0.98 mg g) than in the mid-WAP population (Yelcho) (L: 6.23 ± 0.84%; P: 18.63 ± 1.17%; E: 136.67 ± 7.08 Joules; FA: 10.93 ± 0.63 mg g) and southern population (Rothera) (L: 4.60 ± 0.51%; P: 13.11 ± 0.98%; E: 98.37 ± 5.67 Joules; FA: 7.58 ± 0.48 mg g). We hypothesize these differences in the nutritional condition could be related to a number of biological and environmental characteristics. Our results can be interpreted as a consequence of differences in phenology at each location; differences in somatic and gametogenic growth rhythms. Contrasting environmental conditions throughout the WAP such as seawater temperature, quantity and quality of food from both planktonic and sediment sources, likely have an effect on the metabolism and nutritional intake of this species.

Introduction

Nutritional condition is a key biological response to environmental change, since it is a factor that is expected to limit the capacity of a diverse range of biological mechanisms to respond to biotic and abiotic variability (Somero, Lockwood & Tomanek, 2017). On the one hand, in important aquatic species for aquaculture and fisheries, nutritional condition has generally been analyzed through multiple methods such as DNA:RNA, proximate composition and fatty acids (Tacon & Metian, 2013; Tan et al., 2021). In these studies, nutritional condition has been used as an indicator of food quality for human consumption, since it allows detection of molecules with high nutritional value (Lah et al., 2017; Lorenzo et al., 2021). On the other hand, in ecological studies, nutritional condition has great potential to be used as an indicator of the energy reserves of aquatic organisms (Vesterinen et al., 2020; Bascur et al., 2020). For instance, a recent study on the Antarctic bivalve Nuculana inaequisculpta found differences in the nutritional condition of individuals on a transect away from a retreating glacier in a small WAP fjord (Bascur et al., 2020), and consistent with variation in other population attributes such as genetic diversity (Muñoz-Ramírez et al., 2021). However, understanding of how nutritional condition of different populations of marine invertebrate species vary in response to prevailing environmental conditions at large spatial scales is still very limited in Antarctic ecosystems.

Biochemical composition has been widely investigated in a range of marine bivalve species from high and low latitude ecosystems (e.g., Ahn et al., 2003; Pogoda et al., 2013). Lipids, protein, carbohydrates and fatty acids all have crucial roles in development, metabolism and functioning of marine organisms (Somero, Lockwood & Tomanek, 2017). In species with a wide geographic distribution, these biochemical reserves can be modified by abiotic factors such as sea temperature or food availability (Guzmán-Rivas et al., 2021). However, biotic factors can also play an important role. For example, it has been shown that there is a close relationship between biochemical composition and reproduction in marine invertebrates, since there is a significant expenditure of biochemical reserves in the production of gametes, which are released at the time of spawning (Mathieu & Lubet, 1993; Darriba, Juan & Guerra, 2005; Ngo et al., 2006; Li et al., 2011). In this way, it is necessary to consider reproductive status when the reproductive cycle is not known with certainty, or there is no information available on gonad maturity of the samples. Accordingly, species biochemical composition and energy content can be used as an indicator to compare the nutritional condition among different benthic populations. For this purpose, the WAP constitutes a valuable natural laboratory in which to evaluate biological variables on a large spatial scale, potentially improving understanding of biological patterns across a contrasting environmental gradient (Barnes et al., 2020; Zwerschke et al., 2021).

The study species, Aequiyoldia eightsii (Jay, 1839), is an infaunal bivalve mollusk of the Protobranchia subclass, distributed patchily in Antarctic and sub-Antarctic areas with muddy sediments (Dell, 1990; González-Wevar et al., 2012). A. eightsii is a long-lived species with a maximum lifespan around 60 years (Nolan & Clarke, 1993; Peck & Bullough, 1993; Román-González et al., 2017). This abundant species can be found from intertidal to deep waters, although it is more frequent at depths less than 100 m, with densities of up to 1,540 individual m−2 (Peck & Bullough, 1993). It has been described as an opportunistic species, since it feeds mainly on organic sediment deposits (Zardus, 2002). However, it can modify its feeding habits by ingesting suspended particles when phytoplankton is available (Davenport, 1988a). Recent studies found an endogenous growth rhythm in this species, likely related to reallocation of energy investment towards growth or reproduction (Román-González et al., 2017). A. eightsii shows a 1:1 male: female sex ratio and a lecithotrophic pericalymma larva (Zardus, 2002). At South Orkney Islands (61°S), individuals of this species reach their sexual maturity when shell length is >20 mm (Peck, Colman & Murray, 2000). Further south, at Rothera Station, A. eightsii showed continuous oogenesis throughout the year with spawning in austral winter (Lau et al., 2018).

Clear patterns of sea ice, seawater temperature, primary productivity and other relevant factors as well as biological change along the environmental gradient at the WAP have been extensively reported (Henley et al., 2019; Morley et al., 2020; Rogers et al., 2020). The mean annual sea-ice duration, defined as the mean number of months per year with an ice concentration higher than 50%, is quite different across the WAP (Smith et al., 2012). For example, in the north this sea ice condition lasts on average about 1–2 months per year, while in the middle of the WAP it lasts about 4 months. In contrast, in the south of the WAP this sea ice condition lasts around 5.5 months per year (Smith et al., 2012). In turn, the surface seawater temperature (10 m) during the summer season shows a clear latitudinal gradient along the WAP, with temperatures between 1–1.5 °C in the north, temperatures between 1–1.25 °C in the middle and temperatures between 0.5-0.75 °C in the south of the WAP (Schloss et al., 2012; Cook et al., 2016). Regarding phytoplankton biomass, Kim et al. (2018) also reported contrasting values during the summer along the WAP. Phytoplankton biomass values between 1–2 µg L−1 in the north, biomass values between 2–5 µg L−1 in the middle, and values close to 4.5–6 µg L−1 in the south of the WAP have been observed (Kim et al., 2018).

This research provides information on nutritional condition (biochemical and energy content), a key biological parameter that correlates with the maintenance and growth of the organism. For this purpose, we used A. eigthsii as a study species, an abundant benthic bivalve with a pivotal role as a nutrient recycler (Cattaneo-Vietti et al., 2000; Lovell & Trego, 2003; Gordillo, Malvé & Moran, 2017). Until now, nutritional condition has been unknown in the study species and remains poorly studied in most Antarctic taxa. Specifically, our data provide evidence of spatial variation in the nutritional condition of an Antarctic bivalve at environmentally contrasting locations along the WAP. Furthermore, this study is the first to provide data about the total energy stored in this species, as an important part of the basal energy budget. Our study establishes a starting point for future experimental or in situ studies addressing how marine invertebrates may respond to climate change in the Antarctic ecosystem.

Materials & Methods

Sample collection

To assess nutritional condition of a key benthic species among localities with contrasting environmental conditions in Antarctica, adult individuals of the bivalve mollusk A. eightsii were collected from three roughly equidistant sites along the WAP. Samples were collected during austral summer by SCUBA diving at 10–15 m depth (Fig. 1). The individuals of the O’Higgins (63.3°19′S, 57°53′W; n = 24) and the Rothera stations (67.6°34′S, 68°07′W; n = 15) were collected during January 2018, while the individuals of the Yelcho station (64.9°52′S 63°35′W; n = 19) were collected during March 2017. Unfortunately, it was not logistically possible to obtain samples simultaneously from all three-study sites and the potential implications of this sampling design are discussed. After collection, all the samples were immediately preserved in 99% ethanol and maintained at –80 °C. Then, samples were transported to the UCSC Hydrobiological Resources laboratory at Concepción, Chile and kept under the same conditions until their analysis 4 weeks later. The collection permits were granted by the UK Government for JR17001 and JR18003 expeditions: 31/2017 and S6-2018/01. Also, a permit was granted for collection adjacent to Rothera Research Station: 33/2017.

Figure 1 Map of the A. eightsii sampling along the Western Antarctic Peninsula (WAP). Filled circles indicate the northern, middle, and southern WAP sampling localities: O’Higgins Base (OB), Yelcho (Ye) and Rothera (Ro), respectively.

Dashed arrows represent Southern Ocean currents, modified from Moffat & Meredith (2018): Antarctic Circumpolar Current (ACC), Antarctic Peninsula Coastal Current (APCC), Coastal Current (CC).

Recently, potentially cryptic species have been documented, suggesting two different lineages of A. eightsii along the WAP (González-Wevar et al., 2019). Accordingly, to avoid biases in the biological response, samples from a single lineage have been used in the present study, following genetic analyses (Muñoz-Ramírez et al., 2020).

Shell length and body mass

These data were obtained as previously described in Bascur et al. (2020). Using Vernier calipers with 0.01 mm precision, we determined the individuals’ sizes, measured as the distance between the anterior and posterior edges of the shell (i.e., shell length). To determine the body mass of each individual, the soft tissue was separated from the shells and washed with abundant distilled water on a 0.2 mm sieve in order to remove salt and sediment. Then, samples were frozen at −20 °C for 24 h in independent labeled Eppendorf tubes and subsequently dried for 48 h at −80 °C by sublimation in a lyophilizer (FDU-701; Operon, Gimpo City, South Korea). Finally, using an analytical balance with a sensitivity of 0.1 mg (LA230S; Sartorius, Göttingen, Germany), body mass was determined as the dry mass of the individuals.

Proximate biochemical composition (lipid and protein content)

Following methods described in Bascur et al. (2020), the proximate composition was measured in 20 mg of homogenized dry mass for each individual and expressed in absolute values (mg 20 mg−1), and then calculated in relative values of dry mass for each biochemical component [% dry mass, (DM) = (mg of component × 100)/mg of DM]. In order to improve the performance of the tests, samples were exposed for 15 min at 6 °C in an ultrasonic bath (AC-120H; MRC, Netanya, Israel) with distilled water (protein content) or dichloromethane: methanol (lipid content), and were then analyzed using the methods outlined below.

Lipid content was quantified both in the dry samples and in the ethanol in which the samples were preserved, following the gravimetric method of Folch, Lees & Stanley (1957), modified by Cequier-Sánchez et al. (2008). Each dried sample was homogenized in amber tubes with 5 mL of dichloromethane: methanol (2:1). Then, samples were combined with 4 mL of 0.88% potassium chloride, mixed for 15 s in a vortex (SBS100-2; Select Vortexer) and centrifuged (S-8; Boeco) for 5 min at 6 °C and 1,500 rpm. The precipitate of each sample was transferred to pre-weighed vials and dried through evaporation using ultrapure nitrogen gas (109A YH-1; Glass Col). Total lipid extract obtained by evaporating the solvent was weighed on a precision balance (120A, Precise) and was calculated by subtracting the weight of the empty vial from the weight of the vial with the lipid extract. A similar method was used to obtain the lipid content that potentially was released from the samples into the solvent in which they were preserved (i.e., ethanol). The ethanol from each sample was evaporated, in a previously weighed flask, through a rotary evaporator (RE-2000A; Winkler). Once the solvent has evaporated, the lipid content was obtained by subtracting the weight of the empty flask from the weight of the flask containing the lipid extract. Since we found a very small quantity of lipid in the single ethanol extracts (only 8–10% of the total individual lipid content) we decided to pool the lipid content found in each ethanol sample with the lipid content found in each individual. Finally, lipid extract of each sample was preserved at –80 °C in dichloromethane: methanol (2:1) with butylhydroxytoluene (BHT) as an antioxidant to avoid sample degradation.

Protein content was quantified using a microplate adaptation of the BIO-RAD colorimetric assay of Lowry et al. (1951). This kit included three reagents: S (aqueous solution of sodium dodecyl sulfate), A (alkaline copper tartrate solution) and B (diluted Folin solution). The dry samples of 4 mg for each individual were homogenized in 200 μL of ultrapure water (Mili-Q, Bedford, MA, USA). Then, 5 μL of the mixture was transferred to a 96-well microplate with 200 μL of Reagent B and 25 μL of Reagent A′ (mixture of 20 μL of Reagent S and 1 mL of Reagent A). Subsequently, the samples were shaken for 15 s in a vortex (SBS100-2, Select Vortexer) and incubated in the microplates for 15 min at room temperature. Finally, the absorbance was measured with a spectrophotometer at a wavelength of 750 nm (ELx808; Biotek, Winooski, VT, USA). The concentration of each sample was obtained using a calibration curve for proteins, created by diluting different concentrations of bovine serum albumin (500-0111; Bio-Rad).

Energy content

The energy content (J 20 mg DM−1) was estimated using a bioenergetics equivalent from the biochemical composition data (lipid and protein), as formerly described in Bascur et al. (2020). The bioenergetics equivalents were calculated through conversion coefficients: (a) 1 mg of lipids = 39.54 J, (b) 1 mg of protein = 23.64 J. An approximation of the total energy content for each individual was calculated by adding the energy equivalents of the biochemical composition (total energy = J mg lipid + J mg protein) (Winberg, 1971; Urzúa et al., 2012; Bascur et al., 2017).

Fatty acid composition

Fatty acid profile was determined through standard methods (Urzúa & Anger, 2011; Bascur et al., 2018; Bascur et al., 2020). Specifically, fatty acid methyl esters (FAMEs) were measured after preparation using the sample’s lipid extract. Lipid extracts were esterified at 70 °C for 1 h in a Thermo-Shaker (DBS-001; MRC, Netanya, Israel) using sulfuric acid (1% in methanol) incubations. Then, each sample was vortexed (SBS100-2, Select Vortexer) with 3 mL of n-hexane and centrifuged for 15 s. This process was repeated 3 times and the supernatant was transferred to labeled tubes. Finally, using a nitrogen evaporator (109A YH-1; Glass Col), fatty acids were concentrated. The measurement of FAMEs was performed using a gas chromatograph (Agilent, model 7890A) at set temperature equipped with a DB-225 column (J&W Scientific, 30 m in length, 0.25 internal diameter, and 0.25 µm film). Using chromatography software (Agilent ChemStation, Santa Clara, CA, USA), individual FAMEs were identified by comparison to known standard fatty acids of marine origin (certificate material, Supelco 37 FAME mix 47885-U (Malzahn et al., 2007; Urzúa & Anger, 2011). Each sample was quantified using a calibration curve for fatty acids, diluting different concentrations of Supelco 37 FAME mix standard.

Statistical analysis

Statistical analyses were performed based on standard methods (Sokal & Rohlf, 1995; Clarke & Gorley, 2006; Zuur, Ieno & Graham, 2007) in the STATISTICA V8 and PRIMER V6 (+ PERMANOVA) software packages with a 95% confidence level (p < 0.05). The assumptions of the ANOVA analysis were evaluated with Kolmogorov-Smirnov tests for normality and Levene test for homogeneity of variances. Considering that sample sizes for each group were different, a type 3 sum of squares was used for ANOVA and PERMANOVA analyses. Besides, when significant differences were detected for ANOVA or Kruskal-Wallis test, post hoc Tukey HSD or multiple range tests with a Bonferroni correction were performed to assess differences among localities, respectively. All analyzes were performed with locality as a factor [with 3 levels: O’Higgins station (northern WAP), Yelcho station (middle WAP) and Rothera station (southern WAP)].

The shell length of A. eightsii individuals collected at the three study localities was analyzed through a one-way ANOVA as assumptions of normally distributed data and homogeneity of variances were fulfilled. Because these assumptions (normality and homogeneity) were not fulfilled for soft tissue dry mass, lipid and protein content (mg and %DM) and energy content of A. eightsii individuals captured at the three study localities, these variables were analyzed by non-parametric Kruskal-Wallis tests. Also, the assumptions of normality and homogeneity of variances were evaluated for the quantity of each fatty acid (e.g., C16: 0) and for the total values of each group of fatty acids (e.g., total saturated fatty acids, SFA) among the three study localities. The vast majority of fatty acid comparisons were analyzed with a Kruskal–Wallis test because they did not fulfill ANOVA assumptions. The exceptions analyzed with a one-way ANOVA after a log (x + 1) data transformation, were C18:0 (normality: KS = 0.11, p > 0.20; homogeneity: F = 2.91, p = 0.06), C22:6n-3 (normality: KS = 0.14, p > 0.20; homogeneity: F = 2.50, p = 0.09) and the total of SFA (normality: KS = 0.12, p > 0.20; homogeneity: F = 1.89, p = 0.16). On the other hand, the fatty acids C18:2n-6c and C22:1n-9 were only found in two localities, and as they did not fulfill the assumptions of normality and homogeneity, they were analyzed with a Mann–Whitney U test.

In addition, multivariate analyses were conducted to compare fatty acid composition. A one-way permutational multivariate analysis of variance (PERMANOVA) analysis based on Bray-Curtis similarity and fourth root data transformation was performed to evaluate the complete fatty acids data set. Moreover, a similarity percentage analysis (SIMPER) was carried out to observe the percentage of contribution of each fatty acid to dissimilarity among localities. Last, a principal component analysis (PCA) based on Bray-Curtis similarity and square root data transformation was used to visualize the spatial distribution of data and the fatty acid with the highest contribution at each locality (Pearson Correlation > 0.9).

Results

Shell length and body mass

Shell length (mm ind.−1) showed no significant differences among the three study localities (Fig. 2A, Table S1). In contrast, body mass (mg ind.−1) was significantly different between the study localities, since individuals around O’Higgins and Yelcho stations had a higher body mass than individuals from Rothera station (Fig. 2B, Table S2).

Figure 2 Jitter boxplot of (A) shell length (mm ind.−1) and (B) tissue dry mass (mg ind.−1) of adult individuals of A. eightsii collected from three different localities of the Western Antarctic Peninsula.

Different letters on box indicate significant differences among sites after a multiple range test with a Bonferroni correction. In the boxplot, the horizontal end of the box nearer to zero represents the 25th percentile and the horizontal end of the box more distant from zero represents the 75th percentile. The horizontal black line within the box indicates the median and the red line within the box indicates the mean. Whiskers above and below the box represent 1.5 times the interquartile range from the box, respectively. Black circles above and below the whiskers are outliers (n = 58).

Proximate biochemical composition and energy content

Significant variation among locations were found for lipid content (mg 20 mg DM−1; Fig. 3A, Table S2), lipid percentage (% DM; Fig. 3B, Table S2), protein content (mg 20 mg DM−1; Fig. 3C, Table S2), protein percentage (% DM; Fig. 3D, Table S2) and energy content (J 20 mg DM−1; Fig. 4, Table S2). In all these cases, higher values occurred at O’Higgins station compared to Yelcho and Rothera stations.

Figure 3 Jitter boxplot of (A) lipid content (mg 20 mg−1), (B) lipid content (% DM), (C) protein content (mg 20 mg−1), (D) protein content (% DM) of adult individuals of A. eightsii collected from three different localities of the Western Antarctic Peninsula.

Different letters on box indicate significant differences among sites after a multiple range test with a Bonferroni correction. In the boxplot, the horizontal end of the box nearer to zero represents the 25th percentile and the horizontal end of the box more distant from zero represents the 75th percentile. The horizontal black line within the box indicates the median and the red line within the box indicates the mean. Whiskers above and below the box represent 1.5 times the interquartile range from the box, respectively. Black circles above and below the whiskers are outliers (n = 58).

Figure 4 Jitter boxplot of the energy content (J 20 mg−1) of adult individuals of A. eightsii collected from three different localities of the Western Antarctic Peninsula.

Different letters on box indicate significant differences among sites after a multiple range test with a Bonferroni correction. In the boxplot, the horizontal end of the box nearer to zero represents the 25th percentile and the horizontal end of the box more distant from zero represents the 75th percentile. The horizontal black line within the box indicates the median and the red line within the box indicates the mean. Whiskers above and below the box represent 1.5 times the interquartile range from the box, respectively. Black circles above and below the whiskers are outliers (n = 58).

Fatty acid composition

One-way ANOVA results showed some significant differences among the fatty acid profiles at the three study localities. The amount of total fatty acid, saturated fatty acid, monounsaturated fatty acid, polyunsaturated fatty acid n−6, polyunsaturated fatty acid n−3, and total polyunsaturated fatty acid was higher in O’Higgins station individuals than those from Yelcho and Rothera station (Table 1).

Table 1 Fatty acid composition (expressed in mg of fatty acid g dry mass−1 and in % of total FA pool in parentheses) of the soft tissue of A. eightsii collected from three different localities of the Western Antarctic Peninsula.

	Locality			
Fatty acid	O’Higgins (63°S)	Yelcho (64°S)	Rothera (67°S)	Stat. value	p value	
C11:0	0.24 ± 0.09 (1.47)a	0.13 ± 0.02 (1.19)b	0.19 ± 0.04 (2.51)a	22.03	<0.001	
C12:0	0.29 ± 0.05 (1.78)a	0.25 ± 0.03 (2.29)b	0.25 ± 0.03 (3.30)b	10.22	<0.01	
C13:0	0.26 ± 0.08 (1.59)a	0.16 ± 0.02 (1.46)b	0.27 ± 0.07 (3.56)a	30.21	<0.001	
C14:0	0.65 ± 0.21 (3.98)a	0.48 ± 0.04 (4.39)ab	0.37 ± 0.10 (4.88)b	20.71	<0.001	
C15:0	0.24 ± 0.07 (1.47)a	0.20 ± 0.05 (1.83)a	0.19 ± 0.05 (2.51)a	6.38	<0.05	
C16:0	4.60 ± 1.42 (28.17)a	2.58 ± 0.65 (23.6)b	2.14 ± 0.55 (28.23)b	30.18	<0.001	
C17:0	0.49 ± 0.14 (3.00)a	0.32 ± 0.12 (2.93)b	0.15 ± 0.03 (1.98)c	39.25	<0.001	
C18:0	2.19 ± 0.47 (13.41)a	1.30 ± 0.30 (11.89)b	1.27 ± 0.38 (16.75)b	36.65	<0.001	
C20:0	0.62 ± 0.12 (3.80)	0	0	–	–	
C22:0	0	0.31 ± 0.11 (2.84)	0	–	–	
C23:0	0.43 ± 0.09 (2.63)a	0.27 ± 0.10 (2.47)b	0.34 ± 0.03 (4.49)b	26.39	<0.001	
Total SFA	10.01 ± 1.35 (61.30)a	6.00 ± 0.76 (54.89)b	5.17 ± 0.65 (68.21)b	58.27	<0.001	
C14:1n−5	0.67 ± 0.28 (4.10)a	0.72 ± 0.17 (6.59)a	0.21 ± 0.02 (2.76)b	28.31	<0.001	
C16:1n−9	0.72 ± 0.38 (4.42)a	0.60 ± 0.27 (5.49)a	0.22 ± 0.09 (2.90)b	24.71	<0.001	
C18:1n−9	1.06 ± 0.39 (6.49)a	1.85 ± 0.25 (16.93)b	0.46 ± 0.11 (6.07)c	45.04	<0.001	
C20:1	0.99 ± 0.33 (6.06)	0	0	–	–	
C22:1n−9	0	0.17 ± 0.05 (1.56)a	0.15 ± 0.03 (1.98)a	97.00	0.12	
Total MUFA	3.44 ± 0.42 (21.07)a	3.34 ± 0.72 (30.56)a	1.04 ± 0.17 (13.72)b	32.80	<0.001	
C18:2n−6c	0.32 ± 0.12 (1.96)a	0.21 ± 0.08 (1.92)b	0	94.50	<0.01	
C18:2n−6t	0.24 ± 0.09 (1.47)a	0.14 ± 0.02 (1.28)b	0.15 ± 0.02 (1.98)b	17.90	<0.001	
C18:3n−6	0.36 ± 0.13 (2.20)a	0.21 ± 0.04 (1.92)b	0.14 ± 0.02 (1.85)c	36.61	<0.001	
Total n−6 PUFA	0.92 ± 0.06 (5.63)a	0.56 ± 0.04 (5.12)b	0.29 ± 0.08 (3.83)c	46.29	<0.001	
C20:3n−3	0.43 ± 0.13 (2.63)a	0.25 ± 0.07 (2.29)b	0.14 ± 0.03 (1.85)c	42.81	<0.001	
C20:5n−3	1.03 ± 0.51 (6.31)a	0.42 ± 0.19 (3.84)b	0.59 ± 0.43 (7.78)b	17.69	<0.001	
C22:6n−3	0.50 ± 0.18 (3.06)a	0.36 ± 0.11 (3.29)b	0.35 ± 0.22 (4.62)b	5.33	<0.01	
Total n−3 PUFA	1.96 ± 0.33 (12.00)a	1.03 ± 0.09 (9.43)b	1.08 ± 0.23 (14.25)b	23.85	<0.001	
Total PUFA	2.88 ± 0.28 (17.64)a	1.59 ± 0.10 (14.55)b	1.37 ± 0.21 (18.07)b	32.37	<0.001	
Total FA	16.33 ± 0.98 (100)a	10.93 ± 0.63 (100)b	7.58 ± 0.48 (100)c	41.57	<0.001	
Notes:

Values showing different letters in the superscript (a, b, c) of each fatty acid (FA) indicate significant differences among localities (p < 0.05; parametric post-hoc Tukey HSD or non-parametric multiple range tests). Stat. value represent the statistical value obtained in each analysis (F of ANOVA for C18:0, C22:6n−3 and total SFA; U of Mann–Whitney for C18:2n−6c and C22:1n−9; H of Kruskal–Wallis for all the other comparisons).

Abbreviations are the following: SFA, saturated FA; MUFA, monounsaturated FA; PUFA, polyunsaturated FA; SFA, sum of C11:0, C12:0, C13:0, C14:0, C15:0, C16:0, C17:0, C18:0, C20:0, C22:0 and C23:0; MUFA, sum of C14:1n−5, C16:1n−9, C18:1n−9, C20:1 and C22:1n−9; Total n−6 PUFA, sum of C18:2n−6c, C18:2n−6t and C18:3n−6; Total n−3 PUFA, sum of 20:3n−3, 20:5n−3 and 22:6n−3; Total PUFA, sum of n−3 and n−6 PUFA; Total FA, sum of Total SFA, Total MUFA and Total PUFA.

PERMANOVA analysis, which compares the complete fatty acid profile, showed significant statistical differences among locations (Pseudo-F2, 55 = 206.68; p < 0.001; 9999 permutations; Table S3). Those differences also displayed a clear separation in the spatial distribution among the three populations in the PCA plot (Fig. 5). This is consistent with the SIMPER analysis, since the contribution to the dissimilarity was driven by different fatty acids for each comparison between localities (Table 2).

Figure 5 Principal component analysis (PCA) plot based on Bray–Curtis similarity of fatty acid data of adult individuals of A. eightsii collected from three different localities of the Western Antarctic Peninsula.

Variables (fatty acids) are indicated in the vector plot according to Pearson correlation (>0.9). PC1 axis explained 62.4% and PC2 explained 19.2% of the fatty acid profile between individuals from different localities.

Table 2 Similarity percentage analysis (SIMPER) used to evaluate the contribution of each fatty acid found in A. eightsii individuals collected from three different localities of the Western Antarctic Peninsula (n = 58). The table shows fatty acids that contribute more than 4% to dissimilarity (Contr.%) of each comparison.

Locality	Diss.%	FA	Av.Ab. 1	Av.Ab. 2	Av.Diss.	Diss./SD	Contr.%	Cum.%	
OH vs. Ye	15.83	C20:1	0.99	0	2.97	12.59	18.76	18.76	
		C20:0	0.89	0	2.67	15.97	16.85	35.62	
		C22:0	0	0.74	2.23	11.48	14.07	49.69	
		C22:1n−9	0	0.64	1.94	13.85	12.23	61.92	
		C20:5n−3	0.98	0.79	0.65	1.64	4.13	66.05	
OH vs. Ro	18.23	C20:1	0.99	0	3.20	12.56	17.55	17.55	
		C20:0	0.89	0	2.87	15.40	15.77	33.32	
		C18:2n−6c	0.74	0	2.41	9.85	13.24	46.56	
		C22:1n−9	0	0.62	2.01	21.96	11.02	57.58	
		C16:0	1.45	1.20	0.82	1.81	4.48	62.07	
		C16:1n−9	0.90	0.67	0.74	1.77	4.04	66.11	
Ye vs. Ro	11.33	C22:0	0.74	0	2.54	11.72	22.42	22.42	
		C18:2n−6c	0.67	0	2.28	11.32	20.16	42.57	
		C18:1n−9	1.16	0.82	1.18	5.76	10.44	53.01	
		C14:1n−5	0.92	0.68	0.82	4.05	7.25	60.26	
		C16:1n−9	0.86	0.67	0.67	1.72	5.94	66.20	
		C20:5n−3	0.79	0.84	0.50	1.39	4.42	70.62	
Note:

OH O’Higgins station, Ye Yelcho station, Ro Rothera station, Diss.% percentage dissimilarity of each comparison, FA fatty acid, Av.Ab. average abundance of each fatty acid, Av. Diss. the average similarity that each fatty acid contributes, Diss./SD the proportion of similarity and standard deviation, Contr.% the contribution of each fatty acid to the general dissimilarity, Cum.% General additive dissimilarity.

Discussion

The WAP exhibits the strongest gradient in physical conditions in Antarctica and acute recent environmental change makes it an ideal place to explore and study biological responses to climate. The present study provides the first record of interpopulational variability in the nutritional condition of a marine bivalve species along the WAP. We found that individuals of A. eightsii showed significant differences in biochemical composition between three study localities that are likely to have consequences for the populations. Individuals collected at O’Higgins (the northernmost of our study sites) showed a higher lipid, protein, energy content, and fatty acid levels (SFA, MUFA and PUFA) than individuals collected at Yelcho and Rothera stations. The observed differences in the nutritional condition may be due to each population’s ability to adjust their biochemical composition in response to the prevailing environmental conditions at each site within their broad latitudinal distribution range (Guzmán-Rivas et al., 2021).

The biochemical composition of marine invertebrates is influenced by oceanographic changes exhibited at different latitudes (Guzmán-Rivas et al., 2021). It is possible to observe clear differences of some environmental variables along the latitudinal gradient of the WAP (Rogers et al., 2020). In terms of temperature, the clearest partition is between the northern and the central/southern WAP (Cook et al., 2016). In the northern part, the inflow of water masses from the Weddell Sea can maintain ocean temperatures below 0 °C throughout most of the water column (Moffat & Meredith, 2018). Further south, ocean temperature is dominated by diverse factors (e.g., glacial melt) that result in a water column that is particularly warm at depth (1 °C or higher), capped by a thin, transient, warm layer in summer (Cook et al., 2016). Also, there is significant spatial structure in oceanic primary productivity along the WAP, reflecting combined physical and biogeochemical drivers that include water column structure, upwelling and sea ice seasonality (see Rogers et al., 2020). Marked meridional contrasts are evident during summer, as higher concentrations of chlorophyll-a are present during December–March in the south, while the bloom in northern WAP is more limited to the period December–February (Montes-Hugo et al., 2009; Kim et al., 2018). This regional variability is driven by local environmental settings (Kavanaugh et al., 2015). We suggest that our study species would be likely to present intraspecific variability in nutritional characteristics along the WAP, driven by the environmental gradient and contrasting oceanographic parameters. Such differences could be intensified by the contrasting effects of climate change on marine ecosystems along the WAP (Cook et al., 2016). For this reason, future studies should assess the potential local adaptation of its populations (Sanford & Kelly, 2011; Segovia, González-Wevar & Haye, 2020).

Food available in Antarctic sediments, consumed by detritivoral taxa, contains an important source of organic matter from both planktonic and benthic origin (Glover et al., 2008; Mincks et al., 2008). Nevertheless, the amount of this food is not stable, since spatio-temporal variations have been observed in the amount of food available in Antarctic sediment (Isla et al., 2011), which can be correlated to both ambient light levels in the shallows and the supply of detritus to the sea floor. For instance, sediments with a higher content of lipids and proteins (high food quality) were recorded during the autumn and sediments with a higher content of carbohydrates (low food quality) during spring (Isla et al., 2011). Moreover, recent spatial variability in total organic carbon (TOC), as a proxy of food quantity, was found in sediment along a distance gradient from a WAP deglaciating fjord (Kim, Khim & Ahn, 2021). There was a higher proportion of TOC at more distant sites than at those closer to the glacier front (Kim, Khim & Ahn, 2021). These TOC results are consistent with previous results of the nutritional condition of the bivalve Nuculana inequisculpta at different distances from the glacier in the same Antarctic fjord (Bascur et al., 2020). This research found that individuals captured at the site closest (ca. 670 m) to the glacier front had a poorer nutritional condition, with lower lipid and protein content, than individuals captured at the site further (ca. 2,700 m) from the glacier edge (Bascur et al., 2020). In this context, spatial changes in the food quality and quantity available to A. eightsii could be expected due to the environmental gradient along the WAP. This is a factor that could explain the high variability we found between populations from different geographical regions.

Recently, the reproductive cycle and ontogenetic growth rhythms of A. eightsii have been studied at the WAP (Román-González et al., 2017; Lau et al., 2018). These studies point out that this bivalve can exhibit different growth patterns depending on the allocation of energy resources. This suggests that even some coexisting individuals could be in different stages of their gametogenesis or somatic growth cycle (asynchronous growth). Based on this, it could be hypothesized that our Rothera population, with its lower energy content (Fig. 4), is allocating energy towards somatic growth while the O’Higgins and Yelcho populations, with their higher tissue energy content (Fig. 4), could be allocating more energy to gonad growth. Therefore, based on nutritional condition analysis, the different populations within our study, could present asynchronous ecological characteristics in terms of reproduction, recruitment, and somatic growth along the WAP (Steinberg, 2018). Nevertheless, more studies on growth phenology of different populations of this species along the WAP are necessary in order to support this argument.

Within lipid composition, fatty acids have a pivotal role in the membrane function, nervous system development (Beltz et al., 2007), immune response (Bell et al., 2006; Fritsche, 2006), gonadal maturation (Hurtado et al., 2012; Bolognini et al., 2017), growth (Marshall, McKinley & Pearce, 2010) and as energy sources in long-term starvation conditions (Auerswald et al., 2015). It is thought that most mollusks, including bivalves, lack the capacity to biosynthesize n−3 and n−6 PUFA de novo (Zhukova, 2019). That is, fatty acids such as EPA (eicosapentaenoic acid: C20: 5n−3) and DHA (docosahexaenoic acid: C22: 6n−3) are obtained exclusively through food. In this context, A. eightsii individuals from O’Higgins station had a higher quantity of total fatty acids (especially PUFA as EPA and DHA) than individuals collected at Yelcho and Rothera stations, likely influenced by different food quantity or quality, either in sediment or from phytoplankton (Montes-Hugo et al., 2009; Schofield et al., 2017). Fatty acids can be used as biomarkers of trophic relationships (e.g., see Hughes et al., 2005). Fatty acid markers have proved highly successful in assessing the trophic ecology of Antarctic marine species (e.g., Yang et al., 2016; Servetto et al., 2017; Rossi & Elias-Piera, 2018). Considering the fatty acid profiles found in the present study and the use of fatty acid biomarkers available in the literature, we suggest that A. eightsii has an omnivorous feeding behavior, mainly consuming flagellates, detritus, different types of algae and meiofauna (Table 3). On the other hand, one remarkable result is that individuals at Yelcho had higher levels of detritus biomarkers (C22:0 and C18:1n−9), while individuals at O’Higgins had higher levels of microalgae markers such as diatoms and dinoflagellates (C20:5n−3 and C16:0) (Table 3). While the composition of the phytoplankton species within the bloom is relatively consistent across the WAP, there is up to a 5 fold variation in integrated water column chlorophyll-a from year to year (Schofield et al., 2017). The nature of the bloom is strongly associated with sea ice and is expected to be impacted by ocean warming (Deppeler & Davidson, 2017). In this respect, A. eightsii are well suited for this variability in food supply as they are known to switch between filter and deposit feeding, depending on the availability of phytoplankton, a strategy that has been linked to their continuous oogenesis around Rothera Point (Lau et al., 2018). Such fatty acid and diet profiles represent a valuable contribution to baselines for future studies on WAP marine food webs.

Table 3 Fatty acid biomarkers used for trophic relationships in benthic and pelagic marine environments.

Food source	Fatty acid biomarker	References	
Bacteria in general	Odd numbered SFA	Volkman et al. (1998)	
Detritus	C16:0, C22:0, C18:0 + C18:1n−9	Dalsgaard et al. (2003)	
Green algae	C18:2n−6, C18:3n−6	Cañavate (2018)	
Brown algae	C18:1n−9, C18:2n−6, C20:5n−3, C16:0	Zhukova (2019)	
Phaeocystis	C18:1n−9, C18PUFA + C22:6n−3	Legeżyńska, Kędra & Walkusz (2014)	
Heterotrophic flagellates	C18:2n−6, C22:6n−3	Zhukova (2019)	
Flagellates in general	C18PUFA + C22:6n−3	Legeżyńska, Kędra & Walkusz (2014)	
Red algae	C20:5n−3, C16:0	Legeżyńska, Kędra & Walkusz (2014)	
Meiofauna	C22:6n−3, C18:1n−9	Zhukova (2019)	
Zooplankton (e.g., copepods)	C20:1, C22:1n−9	Kelly & Scheibling (2012)	
Diatoms and dinoflagellates	C22:6n−3, C20:5n−3	Dalsgaard et al. (2003), Cañavate (2018)	
Note:

Abbreviations: SFA, saturated fatty acid; PUFA, polyunsaturated fatty acid.

Limitations and future directions

The absence of information on gonadal maturation or development of the analyzed individuals is considered an important limitation in this study. Our samples were collected in summer, temporally distinct from the spawning season described for A. eightsii in the southern WAP as during winter (Lau et al., 2018). This suggests gonad maturation would form a minor (if any) component of the variation between locations, especially since we are comparing a quite narrow biogeographic range. However, it is necessary to take into account that there could be spatial variation of the reproductive period in this species at different locations in the WAP. In this context, continuous reproductive analysis (i.e., gonadal maturation) using A. eightsii at a number of sites along the WAP environmental gradient should be conducted in future studies, since there is a generalized lack of information on this topic within Antarctic marine invertebrates.

Ideally, future studies should also consider the collection of environmental parameters (e.g., seawater temperature, salinity, etc.) in order to evaluate any potential relationship between biological and environmental data. There are few research centers along the WAP with the capacity to obtain long-term environmental data (e.g., Carlini, Palmer, Rothera). Unfortunately, in the case of the Chilean bases O’Higgins and Yelcho, there are no oceanographic monitoring programs and data could not be taken by other means. For this reason, it was not possible in our study to include environmental data to provide an overall picture at the three study sites. In this context, we emphasize the urgent need to obtain long-term oceanographic data in the northern WAP. In this way, a more representative monitoring of the effect of regional warming on the WAP should improve our understanding of the impacts of climate change on the biology of Antarctic marine invertebrates.

Another consideration is that Yelcho samples were collected eight months earlier than O’Higgins and Rothera samples due to logistical difficulties related to working in isolated and strongly seasonal ecosystems with limited access. This region may experience significant interannual variability, driven by the Southern Annular Mode (SAM) and El Niño-Southern Oscillation (ENSO) (Martinson et al., 2008; Santamaría-del-Ángel et al., 2021), which can translate into biotic variability. In this context, the oceanographic variables such as temperature and salinity at the southern area of Anvers Island (where Yelcho is located) indicated only limited (but significant) interannual variation between the summer seasons of 2017 and 2018 (Fig. S1). On the contrary, chlorophyll-a did not display significant differences between summer seasons of 2017 and 2018 (Fig. S1). Those differences, especially in temperature between the 2 years at Yelcho, while not being lethal to adults, could influence metabolism (e.g., Davenport, 1988b) and therefore the balance between energy gains and costs, modifying energy storage and growth (e.g., Morley et al., 2016). Furthermore, temperature can alter the composition of phytoplankton communities (Schofield et al., 2017) and the nutritional properties of the organic matter stored in the sediment (e.g., Malinverno & Martinez, 2015), causing a change in the type of food available for benthic species. In turn, this limitation also makes it difficult to relate biological aspects to environmental variability, given the lack of information on precise gonadal cycle of Antarctic species. Therefore, differences found in our study might not only be driven by spatial variability, but also by a mixed spatio-temporal variability that should be carefully considered in futures studies.

A final limitation is that we did not analyze glycogen content, even though it is an important body component of bivalves. Glycogen is used mainly as an energy source for oocyte production within the gonads (Mathieu & Lubet, 1993). Thus, by analyzing this component, we would have had insights into the stage of gonadal maturation (e.g., mature or immature stage) of individuals. In this context, despite the fact that proteins, lipids and fatty acids are also an important part of the biochemical composition of organisms, we suggest that related future studies prioritize the evaluation of glycogen content and its relationship with the reproductive cycle of A. eightisii.

In spite of limitations mentioned above, the biochemical and energetic results shown here are within previously published ranges for Antarctic marine invertebrates (Heine et al., 1991; McClintock et al., 1991; McClintock et al., 1992). There is also agreement with the predominance of protein content above lipid content, which in our case was almost three times as much protein (13.11–22.34% DM) as lipid (4.60–8.30% DM). Furthermore, differences were found between the O’Higgins and Rothera samples even though they were captured on exactly the same date. Only the Yelcho data should be interpreted with caution due to the difference in the date of collection, which could potentially be affected by interannual environmental differences. Therefore, we suggest that our study represents a valuable first step, highlighting the importance of evaluating the relationship between physiological and regional oceanographic processes, influencing the nutritional condition of benthic marine invertebrates along the WAP. This will add spatial context to high resolution temporal sampling that is currently undertaken at Rothera (Lau et al., 2018). Additional testing with other taxa and a more comprehensive spatial distribution of study sites can evaluate whether A. eightsii proves to be a good example of how biochemistry of Antarctic marine invertebrates responds to changes in environmental conditions.

Conclusions

The current study provides novel and valuable information on large-scale spatial variation in the biochemical composition and energy content, as a proxy of nutritional condition, of three populations of the bivalve mollusk A. eightsii at the WAP. We observed that the northern population (O’Higgins) had the highest nutritional condition (higher content of lipids, proteins, energy and fatty acids), followed by the middle population (Yelcho), and finally the southern population of the WAP (Rothera) with the poorer nutritional condition (lower content of lipids, proteins, energy and fatty acids). Furthermore, differences regarding feeding biomarkers were also observed between sites with Yelcho individuals having higher levels of detritus biomarkers (C22:0 and C18:1n−9), and O’Higgins individuals having higher levels of microalgae markers. It seems likely that this spatial variability is driven either by different innate growth rhythms of populations or by contrasting environmental conditions (e.g., temperature and food availability) at each study site at the WAP.

Supplemental Information

Supplemental Information 1 Jitter boxplot of (a) seawater temperature (°C), (b) salinity (PSU) and (c) chlorophyll-a (mg/m3) between summer seasons (pooled data January–March) of 2017 and 2018, collected at 0–10 m depth at Palmer station (https://pal.lternet.edu/), southern Anvers I.

Statistical values are given in the right upper corner, after Student t-test (a) and Mann–Whitney test (b and c). In the boxplot, the horizontal end of the box nearer to zero represents the 25th percentile and the horizontal end of the box more distant from zero represents the 75th percentile. The horizontal black line within the box indicates the median and the red line within the box indicates the mean. Whiskers above and below the box represent 1.5 times the interquartile range from the box, respectively. Black circles above and below the whiskers are outliers n = 178.

Click here for additional data file.

Supplemental Information 2 ANOVA table for the shell length of A. eightsii individuals collected in three different localities at the WAP.

Click here for additional data file.

Supplemental Information 3 Statistical summary of Kruskal–Wallis test for dry mass, biochemical composition and energy content of A. eightsii individuals collected in three different localities at the WAP.

When significant differences were found, a multiple range test with a Bonferroni correction was used (*p < 0.01; **p < 0.001).

Click here for additional data file.

Supplemental Information 4 PERMANOVA table for the fatty acid composition of A. eightsii individuals collected in three different localities at the WAP.

Click here for additional data file.

Supplemental Information 5 Shell length, body mass, lipid, protein, fatty acids and energy content data of A. eightsii in three different localities at the WAP.

These data were used for statistical analysis to compare three different populations along the Western Antarctic Peninsula (WAP).

Click here for additional data file.

Supplemental Information 6 Seawater temperature, salinity and chlorophyll-a data collected from South Anvers Island during summers of 2017 and 2018.

These data were used in order to support the discussion argument of the potential effect of interannual variation on the study species biochemical composition at Yelcho station (mid WAP).

Click here for additional data file.

We thank the divers that collected samples at O’Higgins, Yelcho and Rothera stations. Special thanks to Sara García-Ravelo for her valuable input with the English proof-reading and general improvement of this manuscript. Constructive reviews from K. Liversage and other two anonymous reviewers considerably improved this publication.

Additional Information and Declarations

Competing Interests

Author Contributions

Field Study Permissions

Data Availability

The authors declare that they have no competing interests.

Miguel Bascur conceived and designed the experiments, performed the experiments, analyzed the data, prepared figures and/or tables, authored or reviewed drafts of the paper, and approved the final draft.

Simon A. Morley conceived and designed the experiments, authored or reviewed drafts of the paper, and approved the final draft.

Michael P. Meredith analyzed the data, authored or reviewed drafts of the paper, and approved the final draft.

Carlos P. Muñoz-Ramírez conceived and designed the experiments, performed the experiments, analyzed the data, prepared figures and/or tables, authored or reviewed drafts of the paper, and approved the final draft.

David K. A. Barnes conceived and designed the experiments, authored or reviewed drafts of the paper, and approved the final draft.

Irene R. Schloss analyzed the data, authored or reviewed drafts of the paper, and approved the final draft.

Chester J. Sands conceived and designed the experiments, authored or reviewed drafts of the paper, and approved the final draft.

Oscar Schofield analyzed the data, authored or reviewed drafts of the paper, and approved the final draft.

Alejandro Román-Gonzaléz conceived and designed the experiments, analyzed the data, authored or reviewed drafts of the paper, and approved the final draft.

Leyla Cárdenas performed the experiments, authored or reviewed drafts of the paper, and approved the final draft.

Hugh Venables analyzed the data, authored or reviewed drafts of the paper, and approved the final draft.

Antonio Brante conceived and designed the experiments, performed the experiments, authored or reviewed drafts of the paper, and approved the final draft.

Ángel Urzúa conceived and designed the experiments, performed the experiments, authored or reviewed drafts of the paper, and approved the final draft.

The following information was supplied relating to field study approvals (i.e., approving body and any reference numbers):

The collection permits were granted by the UK Government for JR17001 and JR18003 expeditions of the ICEBERGS proyect (N° PII20150078).

The following information was supplied regarding data availability:

All the raw data is available in the Supplemental Files.

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
