# Peer review of "Interpopulational differences in the nutritional condition of Aequiyoldia eightsii (Protobranchia: Nuculanidae) from the Western Antarctic Peninsula during austral summer"

_PeerJ, doi:10.7717/peerj.12679_

## Round 0.1 · original submission · Major Revisions

Dear Dr. Bascur,

Please excuse the delay in the editorial process. A number of potential reviewers have declined my invitation, for several reasons, mostly because they would have not been able to look at this paper within a due time. In such cases, we had to look for alternative referees. As a consequence, the peer review took unfortunately longer than usual.

Your manuscript has been evaluated by three peer reviewers, and the reviewer comments are appended below. Two reviewers think the topic is interesting and represent a significant contribution to this field. However, one reviewer thinks larger number of data should be used to detect any biotic gradient and I agree with him/her.

The major flaws identified include a) a lack of information in the methods and b) the experimental design (different year of sampling and lack of direct measurements of the environmental conditions).

Based on the referees' recommendations, I arrive at this decision: The manuscript does merit publication in PeerJ but it is not acceptable in its current form and needs a major revision based on the reviews and my general editorial comments below. I therefore invite you to resubmit a revised version.

Please carefully consider the comments of the reviewers and provide a point-by-point response which clearly defines the changes made.

Thank you for your patience with the evaluation process and for choosing PeerJ.

I look forward to receiving your revised manuscript.

Yours sincerely,


Blanca Figuerola
* * *
Academic editor
PeerJ

Editor's general remarks

Please soften some of the wording around the analysis, adding more detail in the limitations of the study.

Also have the language in your manuscript checked by a colleague/coauthor or consider seeking out academic editorial services. Also, PeerJ can provide language editing services - please contact us at editorial.support@peerj.com for pricing (be sure to provide your manuscript number and title).

Reviewer 1 ·

Basic reporting

Overall this article is well written, easy to read and follows a logical sequence. It meets the standards of the journal and will be a valuable contribution to this field of research.

The introduction needs to include more information on the specific study sites, since the aim of the manuscript is to look at biochemical condition along an environmental gradient. Some improvements to the statistical analyses and graphs will improve the overall quality of the manuscript. The results and discussion section could be more concisely written and in some cases, statements in the discussion section need more clarification.

I recommend publication after some changes are made to improve the manuscript. These changes are outlined in my specific reviewer comments below.

Experimental design

No additional comments other than those in the reviewers comments below.

Validity of the findings

The overall conclusions made about the data are reasonable, but some additional justification for these conclusions are needed in some cases. This can be achieved by providing additional references and clarifying some of the statements in the discussion, as specified in the reviewer's comments.

Additional comments

No additional comments

Annotated reviews are not available for download in order to protect the identity of reviewers who chose to remain anonymous.

Reviewer 2 ·

Basic reporting

In general the manuscript is well constructed, although it could benefit from proof-reading by one of the English speaking co-authors.

For example, lines 114 starts with the same word as the previous sentence.

Another sentence which did not make sense to me (although I understood what was meant), was on lines 370 - 372. I think this needs rephrasing for clarification.

Lines 442 – 445 were not clear and needs rephrasing. I am not sure the association with Thorsons Rule is necessary, as there is quite a narrow latitudinal range being investigated. It also comes a bit out of the blue, with no prior explanation of what Thorsons Rule is (and its controversies).

Lines 446 – 452 should also be reconsidered, it replicates what is said in the conclusions below, but far less clearly. It either needs rephrasing or scrapping altogether as it perhaps isn’t required.

Other minor wording issues are likely to be corrected at the copy-editing stage.

The references used were appropriate and up to date, and the background to the study was presented clearly. There are an appropriate number of figures, and the structure of the manuscript is regular. The results presented are relevant to the aims of the manuscript, and raw data has been made available as part of the supplementary information.

Experimental design

The manuscripts fits the aims and scopes of the journal, and has a defined and relevant purpose. The introduction gives appropriate background and states how this fills a knowledge gap, with similar work recently being published, although in a different context and species, by the lead author (Bascur et al. 2020). However, the introduction would benefit from stating the significance of this research, why it matters, especially as it may be considered to non-polar researchers as being a rather small biogeographic region.

The methods are clear and reproducible. I do however have a concern regarding the methods used which require clarification, and perhaps warrants mentioning in the manuscript. It is stated in the method that samples were preserved in ethanol at -80, and were collected at different times (2017 and 2018). Detail regarding the time between collection, and when tissue samples were analysed is important here, as it may have a significant impact on the results. Samples for protein and lipid analysis are usually frozen immediately at point of collection – ironically in the Antarctic this is not always possible, due to transportation issues. It is however my understanding that protein and lipid analyses are affected by preservation in ethanol, especially lipids which can become soluble in solvents. Proteins can also denature and precipitate, which may affect the outcome of your protein assay. Please can you provide comments on this, as it may affect the comparability of your results to other studies and may explain some of the variability that you have found? How long was there between sample collection and lab analysis, and could this have affected the outcome if the measured lipids and proteins were degrading in this time?

While it is mentioned in the discussion (see section below), the interannual variability in temperature between the sites should also be better considered, as temperature may well have increased the metabolism of the organisms, and therefore affected the lipid/protein concentrations. Additionally, this change in temperature may well have changed the type of food available to the organism (even if Chl. a concentrations were the same), and therefore affected the energy available in the form of primary production. Without an analysis of the sediments to confirm, it is only speculation to suggest that this did, or did not have an effect on the results, but it cannot be ruled out.

Validity of the findings

Studies of this type are not common in the Antarctic, and often overcome challenges with sample collection and transport to the host institution. The findings do represent an important advance on our knowledge regarding energetics in an important seafloor species

The data have been provided in a format which is easy to reuse if required, and appear statistically sound. The methods appear robust, although I would seek clarification regarding the differential amount of time the samples were stored in solvents before extraction of proteins and lipids, and the effect this may have on total lipid and proteins detected.

With some rephrasing in sections, as stated above, the overall discussion and conclusions are clear and follow a narrative set-out in the introduction. I do have some further comments on specific parts of the manuscript which I outline below.

Additional comments

Figures
You state in your figure legends that your present means and SD, but you present boxplots. Boxplots typically present the median and interquartile ranges (or hinges), with outliers at the extremities. This not an uncommon mistake, but the two are quite different and therefore show different things.

A mean value can be added to the boxplot, or perhaps a different mean and SD plot would be more appropriate as you have statistically compared the means? If keeping the boxplots please change to the correct definitions in the legend, and I would highly recommend adding the data as ‘jitter’ to show the spread of data over the boxplot. This is now considered best practise for the transparency of data, and can help to show how your data is distributed.

Discussion

Line 363 – what is meant by poor(er?) nutritional condition? Lower protein? Lower lipids? Closer site – closer to what? The glacier edge I presume? Just needs a bit of clarification to make it easier to understand what you mean.
Line 418 – you mention here temporal variations in reproduction should be a minor consideration, but there could also be spatial variations between the known reproductive status at Rothera, and unknown status at the other sites. Worth bearing this in mind.
Line 420 – Gonad development can take over 12 months in polar regions, perhaps up to 24 months, so energy allocation to gonad development could be larger than anticipated.
Line 430 – you suggest that temperature may not have a great effect on the species physiology, in relation to the significant difference in interannual temperature variations at the middle site. This is a surprising statement as they are highly adapted to a low temperature environment with very little temperature fluctuation. I strongly suspect this variation did have an effect on the metabolic activity of the animals, and also on the quality of food available at the seafloor and in the sediments.
Line 432 – More information is needed hear regarding glycogen. What would it have shown or not shown? Why is it important?
Line 439 – You mention here you used widely used marine protocols, but there are no example references. This would be very useful, especially with examples of other studies with ethanol stored samples.

·

Basic reporting

I think in the writing for the results there is too much text to deliver what is essentially only a small amount of information. There are many numbers provided which clutter the text and are unnecessarily repeating the same data already provided in the graphs/tables. Overall, the manuscript presents only a small amount of results; below I have provided an example of how the same results information for the entire study could be condensed into only four clear sentences for the results text:

"Shell length did not vary significantly among the study locations (fig. 2a). Significant variation among locations was found for all other variables, being soft tissue dry mass (fig. 2b), lipid content (fig. 3a), lipid proportions (fig. 3b), protein content (fig. 3c), protein percentages (fig. 3d), energy content (fig. 4), total fatty acids, saturated fatty acids, monounsaturated fatty acids, polyunsaturated fatty acids n-6, polyunsaturated fatty acids n-3, and total polyunsaturated fatty acids (Table 1). In all these cases, greater values occurred at O’Higgins station compared to Yelcho and Rothera, except for soft tissue dry mass which was lowest at Rothera but similar between the other two stations. Multivariate fatty acid profiles also differed significantly among locations (Table 2)."

Experimental design

The study aims to perform a basic comparison among three locations in nutritional values of an Antarctic bivalve, using samples taken of 58 bivalve individuals. Potentially, any differences found could be related to differences in environmental conditions.

Ideally, a comparison like this should also involve direct measurements of the environmental conditions at the different locations. Even with such measurements, any relationship discussed between biotic and abiotic variables would only involve speculation in the absence of proper statistics (e.g. correlation analysis) between biotic and environmental data, which was not done.

As large a number of data points as possible should be used with varying environmental conditions to detect effectively any biotic gradient resulting from environmental variability, but in this study only three locations were used (i.e. even if an attempt was made to test biotic and abiotic data correlation, any given test could only have three data points). There was much discussion about a few factors that may be involved in causing spatial variability in nutrients of this species based on previous research, but a huge number of different environmental or other types of variables could be involved; statistical correlations or direct experimental tests would be most useful on such a topic.

Overall, the results from testing biotic differences among locations only really relates to patterns of spatial variability, which is to be expected as a basic type of variability occurring in all ecological systems, and in most cases is treated as random variation occurring alongside the variation of importance for testing some specific ecological hypothesis.

Only when I got to the discussion at line 421 did I pick up that one of the locations was sampled in a different year to the others (Yelcho in 2017, the other two in 2018), which means the spatial variability is mixed up with temporal variability anyway. I do not think it is useful/informative for the location sampled in 2017 to be directly compared to the others, because in the tests as have been done currently, its not possible for the differences found to be related to anything - maybe or maybe not the differences are related to spatial variability, maybe or maybe not to temporal. And any such spatiotemporal variability may or may not be related to warming, glaciers, salinity, latitudinal gradient, gametogenesis etc.

At a minimum, I would say that the locations would need to be sampled at the same time e.g. the same week or month (within the same year), but even this would provide only a relatively uninformative test of basic spatial variability. It would be best to have numerous locations spanning an environmental gradient to allow statistical correlation between the environmental factor of interest and the nutritional status of populations of this species.

Validity of the findings

As stated in my comments about the experimental design, I do not think the findings are valid given that one location was sampled at a different year to the others, thus variability of nutrient status among different locations (independent of variability among different sampling times) has not actually been tested for as stated. The links made in the manuscript text of the biotic data to environmental factors also have limited validity because based on the current data they are only speculation.

Additional comments

Line 40-41: It is confusing that the first sentence of the abstract is about warming (which I guess is referring to climatic warming) but the focus of the study is molluscan nutrient composition. I would start the abstract at the current line 44, and at the end of the abstract add a sentence about how the variability detected may be associated with many environmental processes, among which is climatic warming and effects of marine-terminating glaciers.

line 414: delete "much needed", or provide some text and references to justify why there is a high need for such baselines in this food web relative to the many other types of food webs for which such baselines would also be useful.

---

## Round 0.2 · Minor Revisions

The authors did a great job addressing the referee’s comments and improving the manuscript although some minor revisions are still required.

Please revise the manuscript according to the reviewers' comments.

Reviewer 2 ·

Basic reporting

All previous comments have been addressed and I thank the authors for being thorough. My only, minor suggestion, is a small improvement to the jitter added to the boxplots. By randomly spreading the jitter you can see the distribution of data points so they do not overlap, and also change the size to differentiate from the outliers.

If you have used ggplot, the code is "geom_jitter(width = 0.2, size = 2)". You can also customise the width and size to suit.

Experimental design

I am satisfied that the previously highlighted concerns have been adequately explained or justified. Methods used have been clarified.

Validity of the findings

I am satisfied that the validity of the findings are proportional to the shortcomings, and the conclusions are appropriate.

Additional comments

I would like to thank the authors for the detailed response to the comments, which makes the review process easier. The manuscript as it stands has better readability. I have suggested minor revisions because of my suggested change to the boxplots. However, I will leave this at the editors discretion, and the manuscript is otherwise ready to accept.

·

Basic reporting

line 157: change to "these data were".

line 366: correct the typo here.

Experimental design

The experimental design is basically allowing comparison of the nutritional and morphological characteristics among three different sampling contexts (i.e. two locations sampling at one month/season, and a third sampled at a different month/season). The inferences that can be gained from analysing data from these sampling contexts together are limited, as it is unknown whether differences are from variability that was spatial or temporal or both. Distinguishing and characterising spatiotemporal variability is normally the first step in understanding ecological patterns, which has not been done for the three locations here, rather it has only been shown that some kind of variability overall occurs. However, using analyses like these to test differences among sampling contexts is still technically valid (even if not too rigorous), and the results may be useful for guiding further research that can better characterise exactly what kind of variability is occurring. The limitations and reasons behind them have now been better explained in this revision.

Validity of the findings

no comment

Additional comments

340-342: it would be good for some actual distance measurements to be provided here (I guess these measurements could be made from satellite or aerial imagery).

403-404: maybe change the wording for the argument made here - some simple correlations with environmental data could have been made by just using a thermometer and a refractometer on water from the positions where each sample was collected, without any need for an oceanographic monitoring program.

---

## Round 0.3 · accepted · Accept

Dear Dr. Bascur,

I am pleased to inform you that your paper has been accepted for publication without further changes.

Thank you for submitting your work to PeerJ. We hope you consider us again for future submissions.

Best regards,

Blanca Figuerola
Academic Editor, PeerJ